# Weighted Gene Co-Expression Network Analysis to Explore Hub Genes of Resveratrol Biosynthesis in Exocarp and Mesocarp of ‘Summer Black’ Grape

**DOI:** 10.3390/plants12030578

**Published:** 2023-01-28

**Authors:** Chengyue Li, Lifang Chen, Quan Fan, Pengfei He, Congqiao Wang, Huaxing Huang, Ruyan Huang, Jiaqi Tang, Shehu A. Tadda, Dongliang Qiu, Zhipeng Qiu

**Affiliations:** 1College of Horticulture, Fujian Agriculture and Forestry University, Fuzhou 350002, China; 2Department of Agronomy, Faculty of Agriculture, Federal University Dutsin-Ma, Dutsin-Ma 821101, Nigeria; 3Lunong Agricultural Technology Co., Ltd., Xiamen 361100, China

**Keywords:** grapes, resveratrol biosynthesis, RNA-seq, WGCNA, hub genes

## Abstract

Resveratrol is a polyphenol compound beneficial to human health, and its main source is grapes. In the present study, the molecular regulation of resveratrol biosynthesis in developing grape berries was investigated using weighted gene co-expression network analysis (WGCNA). At the same time, the reason for the resveratrol content difference between grape exocarp (skin) and mesocarp (flesh) was explored. Hub genes (*CHS, STS, F3’5’H, PAL, HCT*) related to resveratrol biosynthesis were screened with Cytoscape software. The expression level of hub genes in the exocarp was significantly higher than that in the mesocarp, and the expressions of the hub genes and the content of resveratrol in exocarp peaked at the maturity stage. While the expression levels of *PAL, CHS* and *STS* in the mesocarp, reached the maximum at the maturity stage, and *F3′5′H* and *HCT* decreased. These hub genes likely play a key role in resveratrol biosynthesis. Kyoto Encyclopedia of Genes and Genomes (KEGG) pathway analysis further indicated that resveratrol biosynthesis was related to flavonoid biosynthesis, phenylalanine metabolism, phenylpropanoid biosynthesis, and stilbene biosynthesis pathways. This study has theoretical significance for exploring genes related to resveratrol biosynthesis in the exocarp and mesocarp of grapes, and provides a theoretical basis for the subsequent function and regulatory mechanism of hub genes.

## 1. Introduction

Resveratrol is the most important stilbene phytoalexin which exists naturally in plants and is conveyed as part of its defense mechanism [1]. It mainly exists in the form of trans in plants. The trans-formed resveratrol is more stable, and its physiological activity is stronger than that of cis-resveratrol [2]. Resveratrol is a non-flavonoid phenolic compound with antioxidant activity, anti-inflammatory, prevention and treatment of cardiovascular disease, antibacterial, antiviral, and other pharmacological effects [3,4,5,6,7,8,9]. In vitro studies showed that resveratrol has a potential antiviral effect against COVID-19 [10,11]. It has a medical effect and has great potential. The more resveratrol to be used in larger-scale clinical trials, the more resveratrol is needed. Grapes have the highest amount of resveratrol in any known plant. Grapes and their derivatives, including fruit juices and wine, are the most important natural sources of resveratrol [12].

The content of stilbene and its morphology in plants that produce stilbene vary strongly due to various environmental factors [13]. The varieties and their developmental stages are the main internal factors affecting the synthesis of resveratrol in grapes [14]. ‘Summer black’ as a triploid, seedless, dark purple variety, is a popular table grape in China [15]. Divided into three distinct stages based on chemical and morphological characteristics, the initial stage of berry growth includes rapid growth and cell division, accumulation of organic acids (mainly malic and tartaric acid) in vacuoles, and synthesis of tannins and hydroxycinnamic acids. During the transition phase, known to viticulturists as veraison, growth declines and the berries begin to soften. There is reduced production of organic acids and synthesis of volatile secondary metabolites that contribute to the typical aroma and flavor of the wine at the ripening stage [16]. Resveratrol can be transported to different tissues to adapt mechanisms responsive to the environment [17]. Grapes have three clear parts: the exocarp, the mesocarp, and the seeds. The exocarp and mesocarp together constitute the development and maturation of the exocarp [18]. During the veraison, berries experience a burst of reactive oxygen species (ROS), as well as a surge in the expression of genes that code for enzymes that produce antioxidants [19]. Grape exocarp and mesocarp have the same heart-protective effects and have comparable antioxidant potential [20,21]. Flavor-related characteristics, such as sugar and acid content, depend primarily on mesocarp [22]. As the fruit ripens, large amounts of sugar in the mesocarp inhibit fungal growth [23].

Resveratrol in grapes mainly exists in leaves and fruits, and grape exocarp is an important source of resveratrol content [24]. Its content is affected by quite a few factors, and varies greatly between different varieties, different times, and different tissue parts of the same plant [25,26,27,28,29]. Resistance genes and pathogenesis-related (PR) genes are dynamically regulated during fruit development, and the transcriptional level of *PR-1* in ‘Norton’ exocarp is significantly increased during ripening. Phenylalanine pathway genes are up-regulated during ‘Norton’ fruit development, and the transcription level of stilbene synthase genes is steadily increased [30].

Resveratrol is synthesized across the phenylalanine metabolic pathway. Phenylalanine ammonia-lyase (PAL), cinnamate 4-hydroxylase (C4H), 4-coumarate: CoA ligase (4CL), chalcone synthase (CHS), stilbene synthase (STS), resveratrol synthase (ST), which are the key enzymes involved in phenylalanine metabolic pathways. Phenylpropane metabolism is based on phenylalanine or tyrosine as the substrate, under the catalysis of phenylalanine ammonia-lyase to produce trans cinnamic acid, cinnamic acid under the action of C4H to produce coumaric acid [31,32,33]. STS is the key enzyme leading to the biosynthesis of resveratrol and stilbene. The expression of the *STS* family in different tissues and fruit development stages in different tissues and fruit development stages is of great significance for studying the content of resveratrol in fruits [34]. STS and CHS belong to the polyketide synthase superfamily, which are the key enzymes in the biosynthesis of flavonoids and stilbene phytoalexin, respectively. Resveratrol and chalcone synthases are related to plant-specific polyketide synthases [35,36,37,38]. Both CHS and STS use p-coumarate CoA as substrate and they are ubiquitous in plants [39]. CHS and STS are plant-specific Polyketide synthase, but STS evolved independently from CHS [40]. It is reported that *Al4CL* and *AlCHS* genes have potential applications in improving the nutritional composition and flavor of wine [41].

Weighted gene co-expression network analysis is a systemic biology approach to describe correlation patterns between genes in microarray samples and it can be used to find modules of highly correlated genes. A dendrogram was created that similarly expressed genes clustered into discrete branches with the most connected nodes or “centres” at the branch tips [42]. Wang et al. identified four highly expressed hub genes in adrenocortical carcinoma using WGCNA, which were inversely correlated with overall survival [43]. Yu et al. used WGCNA to reveal a set of hub genes related to chlorophyll metabolic processes in *Chlorella vulgaris* in response to androstenedione [44]. Cytoscape is open-source software (https://apps.cytoscape.org (accessed on 15 July 2022)) for analyzing and visualizing biological networks. It is capable of importing networks from various sources, and it also allows users to import tabular node data and visualize it on the network [45]. Hub genes are several genes that are highly linked in a co-expression module and are considered biologically important [46]. CytoHubba is a Cytoscape Plug-in for hub object analysis in network biology. It can be used to explore important nodes in biological networks. In addition, CytoHubba will be able to be combined with other plug-ins to develop new profiling protocols to identify hub genes [47].

In this study, we conducted transcriptome sequencing on grape exocarp and mesocarp at different developmental stages and investigated resveratrol hub genes in summer black grape fruits using WGCNA and Cytoscape software.

## 2. Results

### 2.1. Change of Resveratrol Content in Grape Mesocarp and Exocarp

The contents of resveratrol in grape mesocarp and exocarp were significantly different in different stages. However, there was no significant difference in the resveratrol content of the mesocarp in the first four stages (Figure 1A). The content of resveratrol in the mesocarp of berry increased significantly in S5, which reached 0.39 ug/g. The resveratrol content in the exocarp showed no significant difference in S1, S2, and S3. The content of resveratrol in the exocarp reached the maximum at S5 stage, which reached 6.41 ug/g. Under the given conditions, the retention time of resveratrol was about 0.61 min (Figure 1B).

The Illumina HiSeq sequencing platform was used to perform double-terminal sequencing of cDNA libraries of 18 grape exocarp and mesocarp samples from three stages (S1, S3, S5) using second-generation sequencing technology. A total of 128.01 Gb clean data were obtained following quality assessment and data filtering, with more than 92.78% Q30 scores of clean bases, and the mapped ratios were between 81.48% and 91.17% showing that explicit data can be used for subsequent analysis (Appendix A).

### 2.2. Construction of Weighted Gene Co-Expression Network

By obtaining the RNA expression data of S1, S3 and S5 in the mesocarp and exocarp of the ‘summer black’ grape, the co-expression network was constructed on the basis of the optimal soft threshold. The absolute median difference (MAD) method was used to filter the genes with low expressions or small changes between samples, and 21,558 genes were selected for WGCNA analysis. The genes were divided into different modules, and the gene clustering tree was drawn (Figure 2A). The figure can be divided into three parts, from top to bottom. The first part is the phylogenetic clustering tree of genes. The second part is the module color display of corresponding genes. Each color represents a module in the gene co-expression network constructed by WGCNA. The third part shows the correlation between genes and their modules of each trait-related sample. The redder the color, the more positively correlated it is, whereas the opposite is true for blue. Based on the topological overlap matrix, a heat map of the correlation between genes was drawn, where the darker the color, the stronger the interaction between genes (Figure 2B). As shown in the figure (Figure 2C), ten modules (grey60, midnight blue, black, cyan, blue, brown, magenta, purple, light cyan, and red) were identified from WGCNA, each of which contained a different number of genes. Defined with different color codes, the genes in the modules are highly correlated with resveratrol behavior. By observing the correlation between the module and trait, it was found that three modules were significantly related to the resveratrol content, namely magenta (r = 0.77, *p* = 2 × 10^−4^), cyan (r = 0.76, *p* = 2 × 10^−4^), purple (r = 0.61, *p* = 7 × 10^−3^).

### 2.3. Functional and Pathway Enrichment Analysis

KEGG enrichment analysis was performed on magenta, cyan, and purple modules to determine the metabolic pathways related to resveratrol synthesis (Appendix A). Among these pathways, flavonoid biosynthesis, phenylalanine metabolism, phenylpropanoid biosynthesis, and stilbene biosynthesis were the most significantly enriched (Figure 3). It was found that a total of 45 unigenes (Appendix A) were significantly enriched in the flavonoid biosynthesis pathway, among which 25 unigenes encode CHS, and the remaining 20 unigenes encode F3’5’H (flavonoid 3′, 5′-hydroxylase), CYP73A (trans-cinnamate 4-monooxygenase), CHI (chalcone isomerase), F3H (naringenin 3-dioxygenase), PGT1 (phlorizin synthase), HCT (shikimate hydroxyl cinnamoyl transferase). A total of 25 unigenes were significantly enriched in the phenylpropanoid biosynthesis pathway, among which 12 unigenes encode PAL, and the remaining 13 unigenes encode 4CL, COMT (caffeic acid 3-O-methyltransferase), peroxidase, F5H (ferulate-5-hydroxylase), beta-glucosidase, CCR (cinnamoyl-CoA reductase), HCT. It is interesting that a total of 12 unigenes were significantly enriched in the phenylalanine metabolism pathway, and all of them were encoded as PAL. In total, 11 unigenes were significantly enriched in the stilbene biosynthesis pathway. Eight of them encode STS, and three unigenes encode HCT, C4H.

### 2.4. Screening of Related Hub Genes

Using CytoHubba to visualize the interaction network MCC algorithm, the top 10 genes most related to resveratrol content in the three modules were selected (Figure 4A–C). Through homology comparison and gene annotation interaction, 12 hub genes related to resveratrol biosynthesis were obtained (Appendix A). Based on the pathway map analysis of hub genes (Figure 5), among which 2 genes (*VIT_16s0039g01300, VIT_16s0039g01240*) encode to regulate the PAL enzyme, which plays a role in the first stage of resveratrol biosynthesis. 2 genes (*VIT_16s0100g00900, VIT_16s0100g01190*) encode to regulate the STS enzyme, which plays a role in the final step of resveratrol biosynthesis. In the flavonoid pathway, 3 genes (*VIT_16s0100g00930, VIT_16s0100g00780, VIT_16s0100g01000*) were annotated to the CHS enzyme. 4 genes (*VIT_06s0009g02880, VIT_06s0009g02970, VIT_06s0009g02810, VIT_06s0009g02840*) were annotated to the F3′5′H enzyme. 1 gene (*VIT_14s0030g01950*) was annotated to the HCT enzyme. The first step in the phenylpropanoid pathway is the deamination of phenylpropanoid acid to cinnamic acid by PAL. PAL had no significant change between mesocarp and exocarp from S1 to S3 stage, but increased significantly from S3 to S5 stage. STS catalyzes the biosynthesis of the stilbene backbone from three malonyl-CoA and a CoA-ester of a cinnamic acid derivatives in a single reaction. The contents of *STS* and *PAL* were the same, and S5 reached its maximum value at the exocarp and mesocarp, while the contents of S5 at the early stage were little or none. CHS is an important enzyme in flavonoid metabolism and a key point in regulating the biosynthesis of different flavonoid compounds. *CHS* reached its maximum value at the S5 stage in the exocarp with little accumulation and no significant change. *CHS* in mesocarp was almost nonexistent from S1 to S3 stage, but increased significantly in S5 stage. *F3 ‘5’ H* is synthesized by the flavonoid pathway, biosynthetic by catalyzing the hydroxylation of flavonoids at the 3 ‘, and 5’ positions. *F3 ‘5’ H* increased gradually in the exocarp, reached the maximum value in the S5 stage, and showed a decreasing trend in the mesocarp. In the phenylpropane pathway, HCT catalyzes the formation of p-coumadyl CoA into p-coumadyl shikimic acid, followed by the formation of caffeic shikimic acid. Further functionalization was catalyzed by HCT to produce caffeoyl CoA. *HCT* in exocarp and mesocarp had little change in previous periods, while S5 content increased.

### 2.5. Real-Time Quantitative PCR Validation of the Identified Genes

Eight genes were selected from the four pathways enriched by KEGG, among which 4 were hub genes and 4 were highly expressed genes. RT-qPCR was performed in the exocarp and mesocarp of 5 different stages (Figure 4D). The results showed that the 8 genes expressed at the same level and pattern as the FPKM values of transcriptome sequencing, indicating the reliability of the RNA-Seq data. In addition, the gene expression in the fruit exocarp increased significantly during the ripening stage. The mesocarp changes slightly compared to the exocarp.

## 3. Discussion

In this study, 10 gene co-expression modules were obtained by constructing WGCNA, and three co-expression modules magenta, cyan, and purple were identified as resveratrol-specific modules, the genes in these modules were involved in resveratrol biosynthesis during the grape growth process of grapes. The hub genes (*CHS*, *STS*, *F3’5’H*, *PAL*, *HCT*) of resveratrol biosynthesis in mesocarp and exocarp at different developmental stages were further screened by Cytoscape. It was found that flavonoid biosynthesis, phenylpropanoid biosynthesis, phenylalanine metabolism, and stilbene biosynthesis were the most significantly enriched pathways.

Many compounds are derived from phenylalanine via the general phenylpropanoid pathway, and multiple downstream derivatives of the original phenylpropanoid scaffolds are ubiquitous in plants and are used to maintain their structural integrity, UV protection, reproduction, and growth. It plays an important developmental role in cell physiology and signaling. Hu et al. identified four key enzymes (ThPAL, ThC4H, Th4CL, and ThRS) involved in the RES biosynthesis pathway [48]. PAL catalyzes the first reaction in the biosynthesis of various phenylpropanoid natural products from phenylalanine, including lignin, flavonoid pigments, and phytoantitoxins [49]. PAL activity in the exocarp is thought to be directly involved in the synthesis of phenolic substances during grapefruit ripening [50]. The flavonoid metabolism pathway is closely linked to phenylpropanoids. Cinnamyl CoA and 4-cinnamyl CoA coumarin are precursors for flavonoid synthesis. Their biosynthesis is subject to related regulatory enzymes such as PAL in the phenylpropanoid pathway. Subsequently, chalcone is synthesized from 4-chammaryl CoA via CHS. After this step, different flavonoid subsets are controlled by CHS, F3′5′H, HCT, and other enzymes through the production of a modified molecular backbone. Flavonoids and related stilbene all stem from polyketide extension and subsequent cyclization of general phenylpropanoid substrates by CHS or related plant polyketide synthases [51]. Zhu et al. found that *MYB30* and *MYB14* form an inhibitory activation module with *WRKY8* in the flavonoid pathway for stilbene biosynthesis in grapes [52]. According to reports, *STS* has developed from *CHS* several times in its evolution [40]. There is a coregulation between the *STS* and *CHS* genes during grape development, which is consistent with previous studies [53]. *CHS* plays an important role in the accumulation of phenolic substances and total flavonoids in grapefruits [54]. Although *STSs* have been isolated and identified in many species, the gene regulation mechanism of stilbene biosynthesis remains largely unknown [55]. In our study, the contents of *STS* and *CHS* were significantly up-regulated at the maturity stage. This was confirmed by previous studies that the up-regulation of *STSs* and *CHSs* may also up-regulate the content of resveratrol, and that enhanced *STS* expression can produce more resveratrol [35,56,57]. Gatto et al. analyzed the expression profiles of STS and PAL and found that their concentrations increased from variety to maturity, which was consistent with our findings [58]. Ageorges et al. showed that the expression of *F3′5’H* structural genes of the flavonoid pathway, was investigated throughout ripening by real-time RT-PCR [59]. There is evidence that the *HCT* gene relates to shikimic acid-mediated regulation of plant phenol metabolism [60]. Yang, Min et al. showed that 31 unigenes encoded 15 key enzymes, mainly related to flavonoid biosynthesis, and HCT was one of them [61]. Fan S et al. found that the *CtHCT* gene appeared to be associated with abiotic stress response through expression profiling of different flowering stages under light and MeJA treatment [62]. Jim et al. found that *PtoHCT9* and *PtoHCT10* are active in winter and play a role in plant development and response to cold stress [63]. Hoffmann et al. found that HCT genes participate in phenylpropanoid biosynthesis and influence the content of various substances, including flavonoids, phenolic acids, stilbenes, and coumarins [64]. Therefore, different interplay mechanisms of *HCT* and *F3′5’H* might occur in berry mesocarp and exocarp.

In addition, according to RNA-seq and RT-qPCR results, we found that the expression profiles of *CHS, STS, PAL, HCT,* and *F3′5’H* were different in exocarp and mesocarp at different stages. Abiotic stress increased the expression of genes encoding various enzymes of the phenolic biosynthetic pathway [65]. Resveratrol is known as a plant phytoalexin. Activation of pathogen defense in the exocarp of nearly ripe berries appears to be a feature of grape ripening. The exocarp, which is exposed to many external stresses such as fungal infections and ultraviolet radiation, may require more resveratrol protection than the mesocarp. The expression level of hub genes in the exocarp was much higher than that in the mesocarp. The exocarp also acts as an interface to the external environment, protecting internal tissues from pathogens and abiotic stressors [66]. Phenylpropanoids-related transcripts with significant differences between grape exocarp and mesocarp throughout the ripening process. The expression of macromolecule transport-related genes remained highly active in the skin during the whole maturation process. Lijavetzky D et al. found that exocarp ripening can be considered a more complex process due to the wide variety of functions active during dermal maturation [67]. This higher complexity, at least in terms of gene expression, is supported by the identification of more specific exocarp activation processes such as stilbene biosynthesis.

The numbers of up-regulated and down-regulated genes across different stages in mesocarp and exocarp are presented in Appendix A. During the S1 stage, a total of 2523 different expressed genes were identified between the mesocarp and exocarp. During the S3 stage, a total of 3139 different expressed genes were identified between the mesocarp and exocarp. During the S5 stage, a total of 2743 different expressed genes were identified between the mesocarp and exocarp. This indicated that exocarp gene expression was more active.

The content of resveratrol in the exocarp was much higher than that in the mesocarp. The exocarp of grape berries acts as a physical and biochemical barrier to protect the ripe berries from pathogens. Studies have shown that the exocarp becomes more resistant to pathogens to protect the ripe grape berries before they ripen [68,69]. Similarly, the expression level of hub genes in the exocarp was much higher than that in the mesocarp. It indicates that these hub genes directly or indirectly affect resveratrol levels.

## 4. Materials and Methods

### 4.1. Sample Collection

The experimental base is located in Haidi Grape Orchard, Xiamen City, Fujian Province, China. The grape variety “Summer Black” of the same age (6 years) is planted in a facility sheltered from the rain and covered with a transparent plastic polyethylene film. Samples were harvested from November 11th to December 20th, 2021. The fruit was collected from five different developmental stages (Figure 6) according to the BBCH scheme [70]: S1, green fruit stage, day 15 after flowering, begin of berry touch; S2, immature green, berry touch complete; S3, red-appeared, berries begin to brighten in color; S4, full-red, softening of berries; and S5, purple-black, berries ripe for harvest. Three clusters per vine, and 6 vines per stage were randomly picked. Mixed berries selected at random from the top, middle and bottom of each cluster were used for analysis. There was no evidence of stress symptoms or disease in each sample. All samples at the same ripening stage with no mechanical damage were harvested. The mesocarp and exocarp were frozen in liquid nitrogen immediately after separation and stored at −80 °C. Three biological replicates were used for all experiments in each stage.

### 4.2. Extraction and Determination of Resveratrol

Resveratrol extraction was based on a previously published method, with slight modifications [71]. The samples used were randomly selected from each grape lot that had been stored in a −80 °C refrigerator. Exocarp or mesocarp samples (1 g) were ground in liquid nitrogen and then 20 mL of ethanol was added, followed by centrifugation at 10,000 rpm for 10 min. The supernatants were evaporated to dryness by rotary vacuum evaporation at 50 °C. Dried residues were then dissolved in 5 mL ethanol and stored at −40 °C before analysis by UPLC (Ultra Performance Liquid Chromatography). The mixture was filtered with a 0.22 µm syringe filter (Biosharp, Anhui, China). The extracted resveratrol solution was analyzed and quantified by UPLC-chromatography-diode array detector (Waters H-Class TUV QDa (Waters Corp, Milford, MA, USA)), respectively. The column temperature was 30 °C, and the detection wave was 306 nm. Elution was performed using a mobile phase made up of 0.1% (*v*/*v*) formic acid aqueous solution-acetonitrile (35:65) at a flow rate of 0.4 mL·min^−1^. The injection volume was 1 μL. The isocratic gradient run time was 2 min. For ultraviolet spectrophotometer detection, each sample was repeated 3 times, the scanning range was 240~600 nm, the maximum absorption wavelength of resveratrol is 306 nm. Trans-resveratrol standard was purchased from Yuanye, Shanghai, China. Samples were extracted in the dark. Each sample was repeated three times.

### 4.3. RNA Sequencing (RNA-Seq)

Using the Illumina HiSeq sequencing platform from the Biomarker Biotechnology Corporation (Beijing, China), the cDNA library of 18 samples of ‘Summer Black’ grape exocarp and mesocarp in three stages (S1, S3, S5) was sequenced by second-generation sequencing technology. After the sample test is qualified, the library is constructed. Eukaryotic mRNA was enriched with magnetic beads with Oligo (dT), and mRNA was randomly interrupted by the Fragmentation Buffer (YEASEN, Shanghai, China). Using mRNA as a template, the first cDNA strand was synthesized using six-base random hexamers. A second cDNA strand was then synthesized by adding buffer, dNTPs, RNase H, and DNA polymerase I, and the cDNA was purified using AMPure XP beads (YEASEN, Shanghai, China). The purified double-stranded cDNA was then end-repaired, A tail was added, and sequencing connectors were connected. AMPure XP beads were used for segment size selection. Finally, the cDNA library was enriched by PCR. After the construction of the library, the Q-PCR method was used to accurately quantify the effective concentration of the library (effective concentration of the library > 2 nM) to ensure the quality of the library. After a qualified library check, different libraries are pooled according to the target on-machine data volume, and sequenced by the Illumina platform. Clean Data was obtained by filtering the dismounted data, and sequence alignment was performed with the specified reference genome. The Mapped Data obtained is used for library quality assessment such as inserted fragment length test, randomness test, etc.

### 4.4. Weighted Gene Co-Expression Network Analysis and Gene Functional Annotation and Expression Analysis

After obtaining the read counts of genes, the correlation and gene significance of gene expression profiles and intra-module connections were analyzed by the weighted gene co-expression network analysis (WGCNA) software tool in BMKCloud (https://www.biocloud.net/, accessed on 10 July 2022). Firstly, data filtering is carried out to detect and eliminate outliers. Secondly, a scale-free network was obtained by clustering the gene expression level of each sample. The soft threshold power was determined by calculating the correlation value between each gene pair. The minimum number of genes per module is set at 50 genes. Finally, a weighted gene co-expression network was constructed based on the correlation of gene expression levels. The correlation between modules and resveratrol was determined by estimating the module-trait relationship. The Topological overlap measure (TOM) was used to calculate WGCNA to calculate the degree of correlation between genes. The module with the highest weighted correlation coefficient is selected as the module of interest. Compared with KEGG, functional annotation of pathways and related metabolic gene information were obtained. The role of selected genes was determined by functional and pathway enrichment analysis to determine the link between these genes and resveratrol.

### 4.5. Screening of Hub Genes

Based on the analysis of transcriptome results, the reliability of transcriptome results verified by real-time quantitative RT-qPCR, and the CytoHubba Visual Interaction Network MCC algorithm of Cytoscape, the expression changes of the top 10 hub genes related to resveratrol biosynthesis in the exocarp and mesocarp of grapes in five different periods were analyzed, and Gephi (https://gephi.org/) (accessed on 21 August 2022) was used to visualize the module network.

### 4.6. Real-Time Quantitative Reverse Transcription PCR Validation

Total RNAs of exocarp and mesocarp were extracted using RNAprep Pure Plant Plus Kit (TIANGEN, Beijing, China) with three biological replicates. Reactions were set up in Hieff qPCR SYBR Green Master Mix (Yeasen, Shanghai, China) according to the manufacturer’s instructions, with gene-specific primers (0.4 μM) in a final volume of 20 μL. RNA integrity was verified by RNase-free agarose gel electrophoresis. The RNA quantity and concentration were checked using the Quawell Q5000 spectrophotometer (Quawell, Sunnyvale, CA, USA). Thermal cycling conditions involved an initial 95 °C melt (5 min), followed by 40 cycles of 95 °C (10 s) and 60 °C (30 s). Assays were conducted with a C1000 Thermal Cycler fitted with a CFX96 Real-time PCR detection system (BioRad, Hercules, CA, USA), and analyzed using the CFX Manager software (BioRad). The gene primers were designed using Premier 5.0 software. Primers used in this study are listed in Appendix A, using *VvGAPDH* as an internal reference. The data were analyzed by the 2^−ΔΔCt^(−delta delta CT) method.

### 4.7. Statistics Analysis

The results were expressed as the mean ± SD of three biological replicates. One-way ANOVA was used to analyze the significance of different samples in different stages, followed by SPSS 26.0 statistical software Tukey multiple comparison tests (*p* < 0.05). The gene expression level was shown by the log10 fold change value. Origin 2021 and Gephi were applied for the construction of the figures.

## 5. Conclusions

Resveratrol-specific modules were obtained by constructing WGCNA, and the genes in these modules were involved in the biosynthesis of resveratrol during grape growth. The key genes of resveratrol biosynthesis in mesocarp and exocarp at different developmental stages were screened out with Cytoscape software. Hub genes such as *CHS, STS, PAL, F3 ‘5’ H* and *HCT* were identified in the exocarp and mesocarp of grape at different developmental stages, showing that they might have related with biosynthesis of resveratrol. The expression level of hub gene in exocarp was significantly higher than that in mesocarp. The exocarp needs to produce more resveratrol to defend against the environment. These results provide valuable insights into the biosynthesis and molecular biological functions of resveratrol.

## Figures and Tables

**Figure 1 plants-12-00578-f001:**
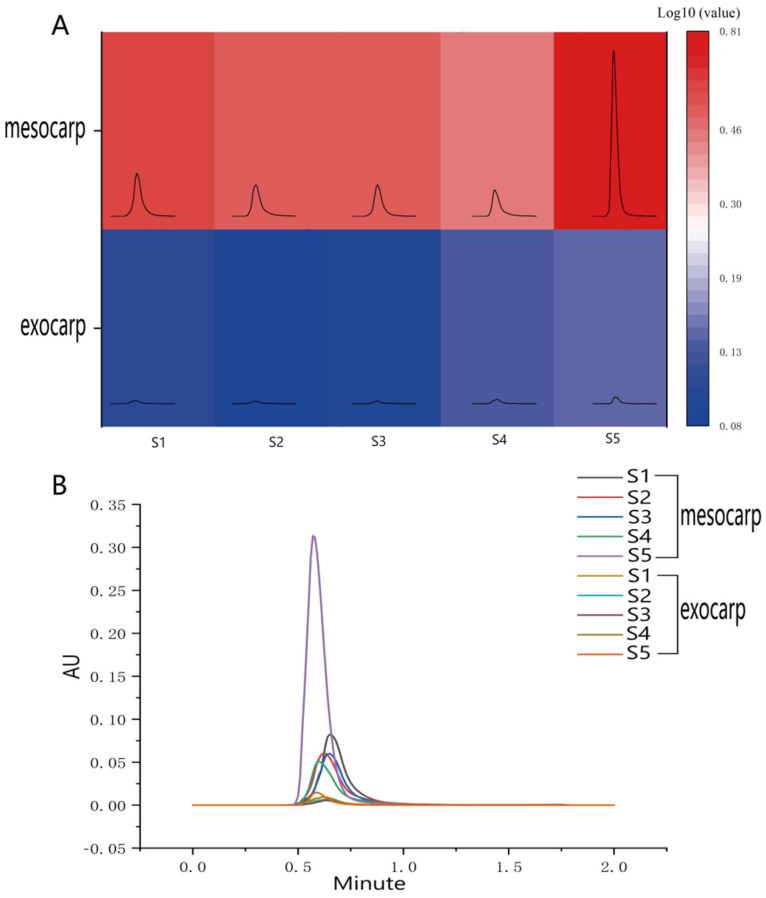
(**A**) UPLC heat map of trans-resveratrol content in exocarp and mesocarp of ‘Summer Black’ grape at different times of berry development, which were expressed as log10 fold change. The darker the heat map, the higher the resveratrol content. (**B**) Ultrahigh performance liquid chromatography retention times of mesocarp and exocarp at different stages of berry. Different line colors represent different stages.

**Figure 2 plants-12-00578-f002:**
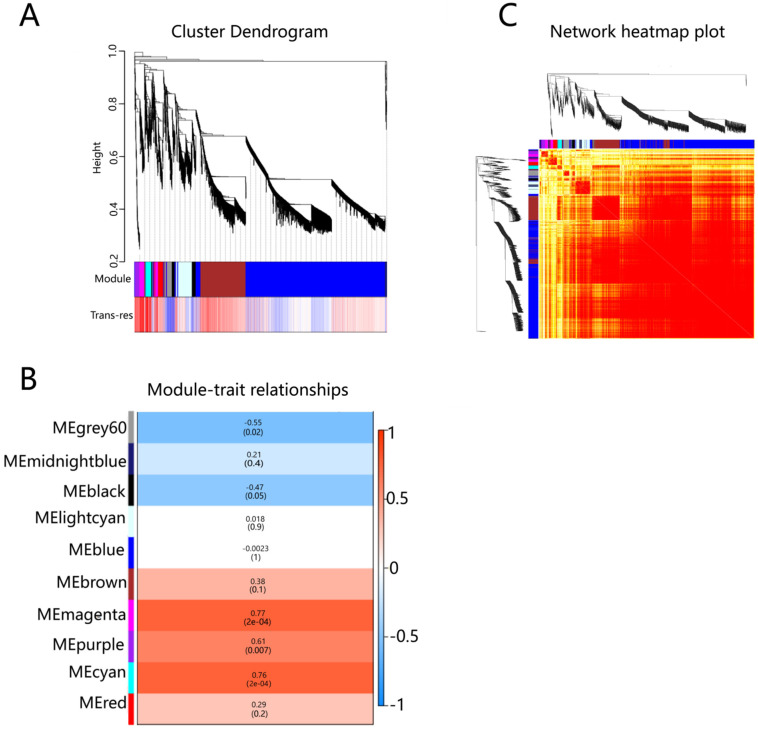
(**A**) Phylogenetic tree heat map associated with trans-resveratrol (trans-res) content. (**B**) Gene co-expression network heat map: on the left and top of the figure are the results of symmetric systematic clustering trees and gene modules. The domain represents the dissimilarity between genes, and the smaller the value, the darker the color. Genes between modules of the same color are darker, and genes between modules are lighter. (**C**) Heat map of correlation between co-expressed gene modules and resveratrol traits. Each row represents a module, and the vertical column represents the content of the resveratrol. The numbers in the rectangular box represent the correlation coefficient between the module and trait, and the corresponding *p* value. Red represents the positive correlation between the module and the trait, and blue represents the negative correlation.

**Figure 3 plants-12-00578-f003:**
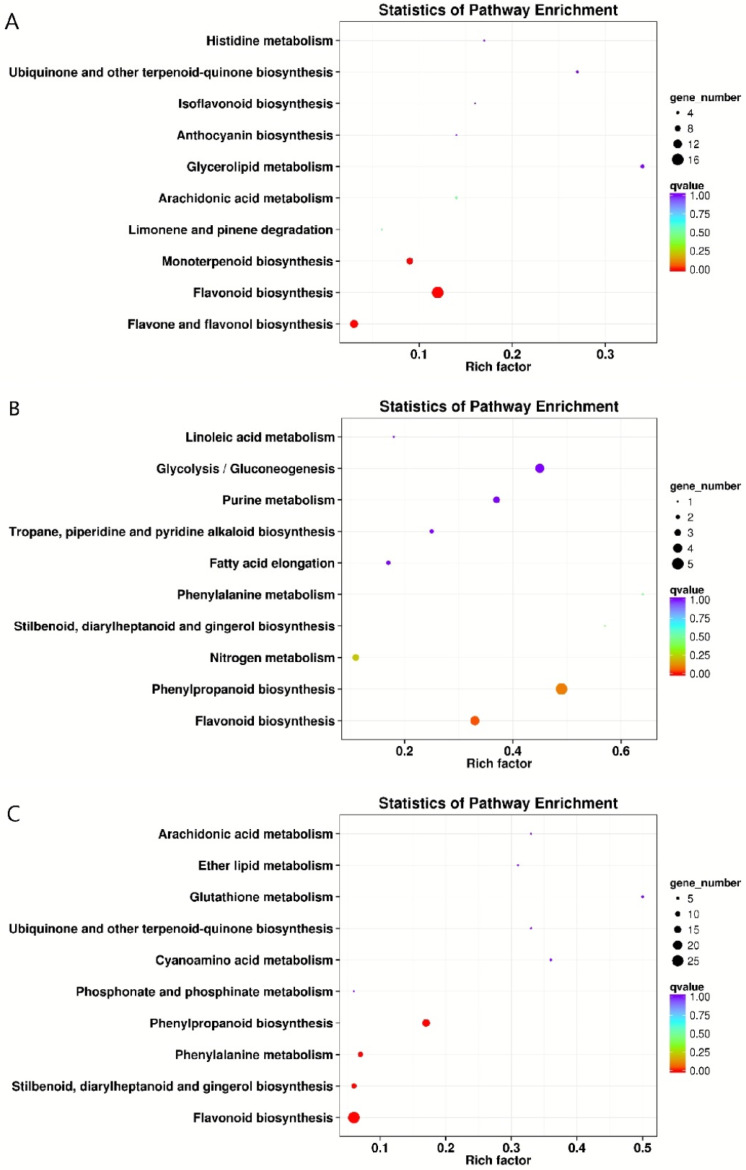
KEGG classification scatter plot of the top 10 pathways with the most enriched genes in the three modules. (**A**) Magenta module. (**B**) Cyan module. (**C**) Purple module. The enrichment KEGG classification is shown on the vertical axis, and the enrichment fraction is shown on the horizontal axis.

**Figure 4 plants-12-00578-f004:**
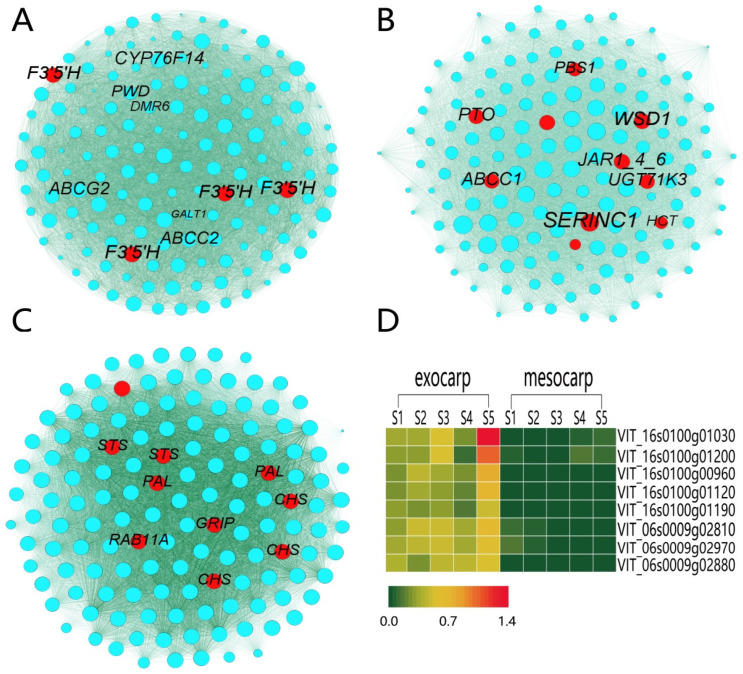
Interaction analysis of hub gene networks in significantly enriched resveratrol modules. (**A**) magenta module, (**B**) cyan module, (**C**) purple module. The hub genes and non-hub genes are represented in red and blue circles, respectively. The size of the point represents the weighted degree, and the larger the weighted degree, the larger the point. (**D**) exocarp and mesocarp quantitative heat map. The expression pattern of 8 selected genes identified by RNA-seq was verified by RT-qPCR.

**Figure 5 plants-12-00578-f005:**
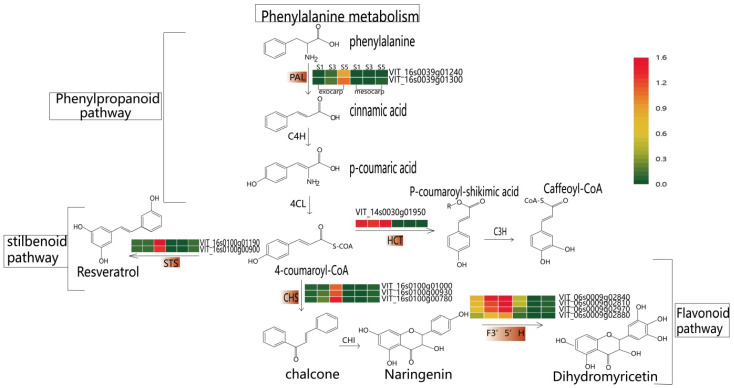
Heat map of pathway hub genes. In the figure, from left to right, changes in the relative expression level of the hub gene in exocarp and mesocarp during S1, S3 and S5 are shown, and the changes are log10 times. The pathway can be divided into four sections: phenylalanine metabolism, phenylpropanoid, stilbene and flavonoid pathways.

**Figure 6 plants-12-00578-f006:**
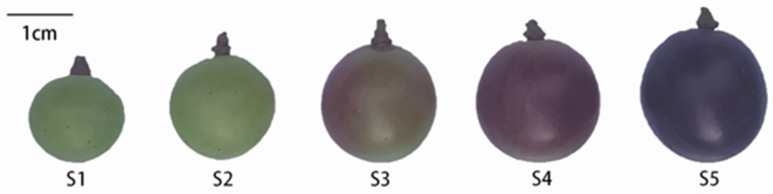
‘Summer black’ grapes at different stages.

## Data Availability

The data presented in this study are available in Appendix A.

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
