# Peer review of "Weighted Gene Co-Expression Network Analysis to Explore Hub Genes of Resveratrol Biosynthesis in Exocarp and Mesocarp of ‘Summer Black’ Grape"

_plants, 2023, doi:10.3390/plants12030578_

Round 1

Reviewer 1 Report (Previous Reviewer 1)

The manuscript is much improved from the previous version and provides an excellent Introduction to the subject and a nice description of development. The Introduction is well written and well referenced.  Grammatical and sentence structure is compromised in a few places, but overall, the manuscript reads well. A few specific comments follow:

Abstract: the abstract needs attention. There are several redundant statements (lines 16-17, 18-19, 23-24). The last statement is unclear: "... provides a theoretical basis and new ideas for improving the content of resveratrol in grapes." Were new ideas presented? and if so, how is that relevant within the context of the paper?

Introduction:

Line 39: I recommend adding "in vitro" to either the beginning or end of the sentence. Otherwise, the statement is misleading. 

Lines 40-42, as stated in my first review, I still do not feel these statements are supported, and I question the relevance of these statements to the subject of the paper.  If not needed to support the work, the authors might consider omission. Commercially-availability RES for chronic pre-clinical and clinical studies is limited, I agree; however, I do not have the sense that the purpose of the paper is to provide new ideas as to increased RES production, as stated in the Abstract. 

M&M's

Lines 117-118: "... bottom half of EACH bunch"? Please review this sentence.

Line 188: is the period after "analyzed" misplaced?

Results: Additional information or further clarification is needed.  Specific comparisons are not clear. Was RES content in exocarp versus mesocarp compared at each developmental stage? Or does this refer to a repeated measures ANOVA in which RES content over time is evaluated between groups. A table may be helpful.

The Discussion is well written and well-referenced.

Conclusion: line 445, please clarify "improvement of its content in grapes." Was an objective of the paper to improve methods to increase RES content in grapes?  The authors might consider omission of this concept throughout the paper, as there is no mention of how RES content in exocarp or mesocarp is increased/improved based on the methods used/demonstrated in the manuscript.

Author Response

Response to Reviewer 1 Comments

Abstract:

Point 1: the abstract needs attention. There are several redundant statements (lines 16-17, 18-19, 23-24). The last statement is unclear: "... provides a theoretical basis and new ideas for improving the content of resveratrol in grapes." Were new ideas presented? and if so, how is that relevant within the context of the paper?

Response 1: We are grateful for the suggestion. The sentence has been rephrased (lines 16-23).

Introduction:

Point 2: Line 39: I recommend adding "in vitro" to either the beginning or end of the sentence. Otherwise, the statement is misleading. 

Response 2: Thank you for this suggested. The sentence has been rephrased (lines 32-33).

Point 3: Lines 40-42, as stated in my first review, I still do not feel these statements are supported, and I question the relevance of these statements to the subject of the paper.  If not needed to support the work, the authors might consider omission. Commercially-availability RES for chronic pre-clinical and clinical studies is limited, I agree; however, I do not have the sense that the purpose of the paper is to provide new ideas as to increased RES production, as stated in the Abstract.

Response 3: Thank you for pointing this out. We have deleted this sentence. In this paper, we explored the genes related to resveratrol biosynthesis in grape exocarp and mesocarp at different developmental stages (lines 21-23).

M&M's:

Point 4: Lines 117-118: "... bottom half of EACH bunch"? Please review this sentence.

Response 4: We modified the sentence. Mixed berries selected at random from the top, middle and bottom of each cluster were used for analysis(lines 102-104).

Point 5: Line 188: is the period after "analyzed" misplaced?

Response 5: Thanks for your kind reminders. The sentence has been rephrased (lines 166-170).

Results:

Point 6: Additional information or further clarification is needed.  Specific comparisons are not clear. Was RES content in exocarp versus mesocarp compared at each developmental stage? Or does this refer to a repeated measures ANOVA in which RES content over time is evaluated between groups. A table may be helpful.

Response 6: Thanks for your kind reminders.Yes, we compared the RES content of exocarp and mesocarp at each developmental stage. We've added the information (Line 192-195).

Conclusion:

Point 7: line 445, please clarify "improvement of its content in grapes." Was an objective of the paper to improve methods to increase RES content in grapes? The authors might consider omission of this concept throughout the paper, as there is no mention of how RES content in exocarp or mesocarp is increased/improved based on the methods used/demonstrated in the manuscript.

Response 7: Thank you for underlining this deficiency. The sentence has been rephrased (lines 425-430).

Reviewer 2 Report (New Reviewer)

The manuscript, entitled " Weighted Gene Co-expression Network Analysis to Explore Hub Genes of Resveratrol Biosynthesis in Exocarp and Mesocarp of 'Summer Black' Grape." This work is merited for publication in Plants after some major modification. So, I have some points that may help to improve the work as follows:

1-Abstract is good but need more explain about the main aim of work

2- The introduction should be extended to discuss the hypothesis and research questions in details. Additionally, the introduction should cover the recent literature related to this subject.

3- Material and methods

The methodologies should be explained in details so that the results are reproducible.

4-Results

The results are clear and important.

5-Discussion
The discussion section still needs improvement, and should be linked to the findings of the previous reports on this topic.

6- The conclusion

A section for conclusions need more explain and should include the most significant findings and future works only.

7- English writing should be checked by a native English-speaking expert.

Author Response

Response to Reviewer 2 Comments

Abstract:

Point 1: Abstract is good but need more explain about the main aim of work.

Response 1: Thank you for this suggested. We've added the information (Line 15-23).

Point 2: The introduction should be extended to discuss the hypothesis and research questions in details. Additionally, the introduction should cover the recent literature related to this subject.

Response 2: Thank you for this suggested. We added the information in the introduction (Line 38-39,47-48,75-76).

Material and methods

Point 3:The methodologies should be explained in details so that the results are reproducible.

Response 3: Thank you for this suggested. We added the information in the Material and Methods

 (Line 166-170, 183-186).

Discussion

Point 4: The discussion section still needs improvement, and should be linked to the findings of the previous reports on this topic.

Response 4: Thank you for this suggested. We added the information in the Discussion

 (Line359-360, 376-382, 417-418).

Conclusion

Point 5: A section for conclusions need more explain and should include the most significant findings and future works only.

Response 5: Thank you for your significant reminding. We added the information in the Conclusion (Line 425-430).

Point 6: English writing should be checked by a native English-speaking expert.

Response 6: Thanks for your kind reminders. We corrected spelling and grammar errors and polished the entire article.

Round 2

Reviewer 2 Report (New Reviewer)

Authors revised Manuscript carefully. The quality of revised Manuscript has been improved as per journal standard. I am recommending for publication.

This manuscript is a resubmission of an earlier submission. The following is a list of the peer review reports and author responses from that submission.

Round 1

Reviewer 1 Report

Plants-2062344

Weighted Gene Co-expression Network Analysis to Explore Hub Genes of Resveratrol Biosynthesis in Exocarp and Mesocarp of 'Summer Black' Grape

Introduction:

The objectives of the study need to be further characterized in the Introduction. Although the final sentence states the study explored hub genes of resveratrol in a specific grape, there is no context for this statement. The differences explored in the paper (e.g., exocarp vs mesocarp, developmental stages, etc.) are not clearly stated in the Intro. Are the authors are attempting to identify biological pathways involved in RES synthesis through gene expression analysis?  Spelling and grammar need minor adjustments throughout the manuscript. 

Lines 36-37: This statement is not sufficiently supported. One of the two citations present is an in vitro study which should be explicitly stated. The authors might consider “potential” antiviral effects here.

Line 42: Do different varieties have different developmental stages? Consider listing the specific developmental stages evaluated in the experiment.

Line 60: Is plant-specific the correct term? Truncated sentence.

M&M’s:

Lines 87-89: Are the developmental stages defined elsewhere or were these 5 stages defined by the authors. If the latter, was color the only factor in developmental stage? Was size a determinant in developmental stage? What is the significance of these 5 stages (as opposed to 3, per se?).

Line 90: Was there a defined minimum # of fruits contained within each experimental “bunch?”

 Fig 1: Is the term “times” referring to “developmental stages?” Ensure consistency with terminology.

Section 2.2: What samples were analyzed? All fruits from a single bunch analyzed separately, or were fruits from a bunch pooled? Are there any descriptive statistics regarding the number of samples collected, # of samples analyzed, etc.?

Section 2.7 Statistics Analysis:  The authors do not provide sufficient detail regarding the endpoints that were analyzed statistically. There is no information regarding what comparisons were made in this experiment.   

Results:

Line 149: Again, what is the significance of having the five developmental stages? Why did the authors expect a difference across the five classifications? Is this just based on color alone?  (is resveratrol related to pigment?).

Line 152: what about S4?

Fig 2, Line 159: Is this the first mention of pericarp? If so, please explain why pericarp is suddenly mentioned. Ensure consistency with terminology.

Fig 3, Line 162: The use of the term, “”choice” is unclear based on the context of this statement. Please clarify.  Mean connectivity may need further clarification as to the relationship w/ scale independence. With higher correlation, there is lower connectivity? Is this a typical relationship one would expect here? Is this figure more representative of methods used as opposed to Results? Reconsider value added.

Lines 180-199: Consider further clarification of the term, “Module.”

Line 224: “Screened out” suggests exclusion.  Is this intentional?

Figure 5: More discussion is needed in relation to the heatmaps generated in this figure. What are some other developmental changes occurring during S5 that could explain an increasing trend? Correlation is not cause/effect.

Figure 6: More discussion is needed in relation to the heatmaps generated in this figure. It is unclear how the pathways shown in the figure relate to each other. Why are only three developmental stages shown?

Line 298-299: There is no mention of temperature measurements in this experiment.  Is temperature considered an abiotic stress in this study?  Unless temperature extremes (outside of normal temperature fluctuations) were demonstrated, this a is highly speculative statement.

Lines 309-310: Conflicting statement? The authors state gene expression did not change, but also states a maximum was reached at S5. Please clarify.

Line 311: It is not clear what “opposite” refers to. A decline in gene expression of HCT and F3’5’H?

Lines 312-315: It is not clear how these two statements are related. Please clarify.   

Lines 318-319. A direct association between hub gene expression and resveratrol was not established in exocarp or mesocarp in this study.  An indirect correlation (not an effect) is unclear.  If all 5 hub genes indirectly “affected” resveratrol levels, a similar trend might be expected in the exocarp and the mesocarp (regardless of the levels of RES and/or gene).  In the exocarp, all five hub genes increased with increased RES levels, whereas in the mesocarp, increases were only seen in 3 of the five genes. The other two genes, HCT and F3’5’H, decreased with resveratrol levels. It is possible these two genes are unrelated to resveratrol. Rather, they may be related to developmental changes (e.g., pigment production) that do not take place in the mesocarp. 

Overall, the manuscript was well-written and well-organized, with only minor grammatical and spelling errors that are easily overcome. Greater emphasis is needed to highlight the objectives of the study in the Introduction, providing greater clarification on the classification of the five developmental stages used throughout the experiment.

Author Response

Point 1: Unfortunately, the novelty of the study is doubted. As far as I understand, all genes identified as hubs in GRN of resveratrol accumulation were described in previous studies, and it is unclear what this study adds to the knowledge.

Response 1: Thank you for this valuable feedback. And realized that these differences might not have been expressed clearly enough in previous manuscripts. We have made improvements to the original manuscript. Our grapes are winter ripening, and there are not many articles on the dynamic changes of resveratrol in winter. In my study, WGCNA and Cytoscape have been used to identify genes related to resveratrol biosynthesis in five stages of grape from berry growth to harvest. In addition, we divided grapes into mesocarp and exocarp to study the changes in resveratrol content and the changes in hub genes respectively.

Point 2: Moreover, the description of methods lacks many important parts. RNA extraction protocol should be described, as well as library construction protocol, sequencing details (instrument, layout, read length), primary data analysis (read quality assessment, trimming, mapping and read per gene counting).

Response 2: Thanks for your kind reminders. We have added details about RNA extraction , library construction and sequencing details (Line 174-184,Table S5).

Point 3: The lack of information affects all the manuscript. The description of weighted gene co-expression network construction is incomplete. Line 182 states that genes with  low expression levels were removed from analysis, but there is no indication of which genes were considered low-expressed. Line 183 states that for GRN construction 21558 genes correlated with the resveratrol content change were selected. But it is almost all grape genes, are they really all correlated with resveratrol content?

Response 3: Thanks for your kind reminders. We added the details of weighted gene co-expression network construction to the methods (Line 156-163). The absolute median difference (MAD) method was used to filter the genes with low expressions and small changes between samples.This sentence was modified according to the comment (Line 243-244).

Point 4: Figure 4 represents KEGG pathways enriched in GRN modules associated with resveratrol content. The figure is of poor quality and it is hard to distinguish between colors. Are the terms with q-value close to 1 represented on the figure? It is strange, because such term enrichment is not statistically significant.

Response 4: Thank you so much for your careful check.The q-value of the KEGG bubble map is not considered significant if it is greater than 0.05. In the figure, the q-value of phenylalanine metabolism, phenylpropanoid, stilbene and the flavonoid pathways is less than 0.05. We replaced Figure 4 with a 300dpi image.

Point 5: Figure 3C and 3D seem to be mixed up.

Response 5: Thanks for your kind reminders. We reformatted the figure.

Point 6: There are several sentences which are really hard to understand, e.g. lines 186-188: "The map is divided into three parts: the first part is the phylogenetic clustering tree of genes, the second part is the module color display of corresponding genes, and the third part shows the correlation between genes and their modules of each trait related sample."

Response 6: I'm sorry that this part was not clearly written in the manuscript. We changed the sentence based on the picture (246-249).

Point 7: In the end I should note that genes encoded actin are not suitable for real-time qPCR experiments, as they are highly unstable (https://pubmed.ncbi.nlm.nih.gov/16166256/). The minimal number of reference genes is two, preferably three, and grape orthologs of truly stable Arabidopsis genes should be used (together with control of their stability using grape expression browsers).

Response 7: Thank you for your significant reminding. We re-selected VvGAPDH as the internal reference gene. The expression levels and patterns of these eight genes were the same as the FPKM values of transcriptome sequencing, indicating the reliability of RNA-Seq. The results of VvGAPDH and Actin as internal reference genes were almost identical.

Reviewer 2 Report

The complexity of plant metabolites is enormous and metabolomics studies is an important field in plant biology. The study of Dr. Chengyue Li and colleagues  describes the regulation of synthesis of resveratrol - a plant metabolite with multiple beneficial effects.

Unfortunately, the novelty of the study is doubted. As far as I understand, all genes identified as hubs in GRN of resveratrol accumulation were described in previous studies, and it is unclear what this study adds to the knowledge.

Moreover, the description of methods lacks many important parts. RNA extraction protocol should be described, as well as  library construction protocol, sequencing details (instrument, layout, read length), primary data analysis (read quality assessment, trimming, mapping and read per gene counting).

The lack of information affects all the manuscript. The description of weighted gene co-expression network construction is incomplete. Line 182 states that genes with  low expression levels were removed from analysis, but there is no indication of which genes were considered low-expressed. Line 183 states that for GRN construction 21558 genes correlated with the resveratrol content change were selected. But it is almost all grape genes, are they really all correlated with resveratrol content?

Figure 4 represents KEGG pathways enriched in GRN modules associated with resveratrol content. The figure is of poor quality and it is hard to distinguish between colors. Are the terms with q-value close to 1 represented on the figure? It is strange, because such term enrichment is not statistically significant.

Figure 3C and 3D seem to be mixed up.

There are several sentences which are really hard to understand, e.g. lines 186-188: "The map is divided into three parts: the first part is the phylogenetic clustering tree of genes, the second part is the module color display of corresponding genes, and the third part shows the correlation between genes and their modules of each trait related sample."

In the end I should note that genes encoded actin are not suitable for real-time qPCR experiments, as they are highly unstable (https://pubmed.ncbi.nlm.nih.gov/16166256/). The minimal number of reference genes is two, preferably three, and grape orthologs of truly stable Arabidopsis genes should be used (together with control of their stability using grape expression browsers).

Author Response

Point 1: Unfortunately, the novelty of the study is doubted. As far as I understand, all genes identified as hubs in GRN of resveratrol accumulation were described in previous studies, and it is unclear what this study adds to the knowledge.

Response 1: Thank you for this valuable feedback. And realized that these differences might not have been expressed clearly enough in previous manuscripts. We have made improvements to the original manuscript. Our grapes are winter ripening, and there are not many articles on the dynamic changes of resveratrol in winter. In my study, WGCNA and Cytoscape have been used to identify genes related to resveratrol biosynthesis in five stages of grape from berry growth to harvest. In addition, we divided grapes into mesocarp and exocarp to study the changes in resveratrol content and the changes in key genes respectively.

Point 2: Moreover, the description of methods lacks many important parts. RNA extraction protocol should be described, as well as library construction protocol, sequencing details (instrument, layout, read length), primary data analysis (read quality assessment, trimming, mapping and read per gene counting).

Response 2: Thanks for your kind reminders. We have added details about RNA extraction, library construction and sequencing details (Line 174-184, Table S5).

Point 3: The lack of information affects all the manuscript. The description of weighted gene co-expression network construction is incomplete. Line 182 states that genes with  low expression levels were removed from analysis, but there is no indication of which genes were considered low-expressed. Line 183 states that for GRN construction 21558 genes correlated with the resveratrol content change were selected. But it is almost all grape genes, are they really all correlated with resveratrol content?

Response 3: Thanks for your kind reminders. We added the details of weighted gene co-expression network construction to the methods (Line 156-163). The absolute median difference (MAD) method was used to filter the genes with low expressions and small changes between samples.This sentence was modified according to the comment (Line 243-244).

Point 4: Figure 4 represents KEGG pathways enriched in GRN modules associated with resveratrol content. The figure is of poor quality and it is hard to distinguish between colors. Are the terms with q-value close to 1 represented on the figure? It is strange, because such term enrichment is not statistically significant.

Response 4: Thank you so much for your careful check.The q-value of the KEGG bubble map is not considered significant if it is greater than 0.05. In the figure, the q-value of phenylalanine metabolism, phenylpropanoid, stilbene and the flavonoid pathways is less than 0.05. We replaced Figure 4 with a 300dpi image.

Point 5: Figure 3C and 3D seem to be mixed up.

Response 5: Thanks for your kind reminders. We reformatted the figure.

Point 6: There are several sentences which are really hard to understand, e.g. lines 186-188: "The map is divided into three parts: the first part is the phylogenetic clustering tree of genes, the second part is the module color display of corresponding genes, and the third part shows the correlation between genes and their modules of each trait related sample."

Response 6: I'm sorry that this part was not clearly written in the manuscript. We changed the sentence based on the picture (246-249).

Point 7: In the end I should note that genes encoded actin are not suitable for real-time qPCR experiments, as they are highly unstable (https://pubmed.ncbi.nlm.nih.gov/16166256/). The minimal number of reference genes is two, preferably three, and grape orthologs of truly stable Arabidopsis genes should be used (together with control of their stability using grape expression browsers).

Response 7: Thank you for your significant reminding. We re-selected VvGAPDH as the internal reference gene. The expression levels and patterns of these eight genes were the same as the FPKM values of transcriptome sequencing, indicating the reliability of RNA-Seq. The results of VvGAPDH and Actin as internal reference genes were almost identical.
